# Research to evaluate safety and impact of long COVID intervention with Ensitrelvir for National Cohort (RESILIENCE Study): A protocol for a randomized, double-blind, placebo-controlled trial

Keiji Konishi [1,2,3*], Shungo Yamamoto [2,4,5], Ryuichi Minoda Sada [2,4,5], Kento Asano [6], Daisuke Onozuka [1,2], Shintaro Tanaka [7], Shogo Miyazawa [8], Masahiro Kinoshita [9], Satoshi Kutsuna [1,2,4,5]

1 Department of Post-infectious disease Therapeutics, Graduate School of Medicine, The University of Osaka, Osaka, Japan, 2 Department of Infection Control and Prevention, Graduate School of Medicine, The University of Osaka, Osaka, Japan, 3 AIDS Medical Center, NHO Osaka National Hospital, Osaka, Japan, 4 Division of Fostering Required Medical Human Resources, Center for Infectious Disease Education and Research (CiDER), The University of Osaka, Osaka, Japan, 5 Department of Transformative Protection to Infectious Disease, Graduate School of Medicine, The University of Osaka, Osaka, Japan, 6 Academic Clinical Research Center, Department of Medical Innovation, The University of Osaka Hospital, Osaka, Japan, 7 Medical Affairs Department, Shionogi and Co., Ltd., Tokyo, Japan, 8 Data Science Department, Shionogi and Co., Ltd., Osaka, Japan, 9 Medical Affairs Department, Shionogi and Co., Ltd., Osaka, Japan

* konishi@hp-infect.med.osaka-u.ac.jp

## Abstract

Long COVID, characterized by persistent symptoms following acute coronavirus disease (COVID-19), presents a substantial challenge to global health. Ensitrelvir fumarate, a novel oral antiviral agent, has demonstrated efficacy in treating of mild-to-moderate COVID-19. This study aims to evaluate the efficacy and safety of ensitrelvir in the prevention of Long COVID in patients without risk factors for severe COVID-19. This multi-center, randomized, double-blind, placebo-controlled trial adopts a decentralized clinical trial design, with participants recruited through partner healthcare institutions. Adults with mild COVID-19 will be randomized within 72 hours of symptom onset in a ratio of 1:1 to receive ensitrelvir (375 mg on day 1, followed by 125 mg once daily on days 2–5) or placebo. The primary efficacy endpoint is the proportion of patients with either "symptoms of fatigue, shortness of breath, difficulty breathing, or disturbances in smell or taste at 1 and 3 months post-treatment initiation" or "symptoms of difficulty with concentration and thinking, difficulty of reasoning and solving problems, or memory loss at 3 months post-treatment initiation". Secondary endpoints include combinations of various symptoms, quality of life, and work productivity. The target sample size is 2,000 participants. This trial will generate data on the potential of ensitrelvir to prevent Long COVID, with important implications for

**Data availability statement:** No datasets were generated or analysed during the current study. All relevant data from this study will be made available upon study completion.

**Funding:** This study is funded by Shionogi & Co., Ltd. The funder was involved in the study design, data analysis, and manuscript preparation, but had no role in data collection or the decision to submit the manuscript for publication. This study is funded by Shionogi & Co., Ltd. Author employees of Shionogi & Co., Ltd. were involved in the study design, and manuscript preparation. Author employees of Shionogi & Co., Ltd. will not be involved in data collection and data analysis.

**Competing interests:** Keiji Konishi, Daisuke Onozuka, and Satoshi Kutsuna have received research support from Shionogi & Co., Ltd., Tokyo, Japan. Shungo Yamamoto, Ryuichi Minoda Sada, Kento Asano declare no competing interests. Shintaro Tanaka, Shogo Miyazawa, and Masahiro Kinoshita are full-time employees of Shionogi & Co., Ltd., and may hold stock in the company. This does not alter our adherence to PLOS ONE policies on sharing data and materials.

**Abbreviations:** COVID-19: Coronavirus disease; SARS-CoV-2: Severe acute respiratory syndrome coronavirus 2;PASC: Post-acute sequelae of COVID-19; DCT: Decentralized clinical trial; IL-6R: Interleukin-6 receptor; EQ-5D-5L: EuroQol 5-dimension 5-level scale; AEs: Adverse events; SAEs: Serious Adverse events.

patient care and public health. The decentralized design enables efficient data collection and minimizes participant burden.

## Trial registration

This study was registered with the Japan Registry of Clinical Trials on February 16, 2024 (jRCTs051230184, https://jrct.mhlw.go.jp/en-latest-detail/jRCTs051230184).

## Introduction

The coronavirus disease (COVID-19) pandemic, caused by severe acute respiratory syndrome coronavirus 2 (SARS-CoV-2), has resulted in unprecedented global morbidity and mortality. Although acute COVID-19 symptoms are well characterized, the long-term consequences, which have collectively been termed as Long COVID or post-acute sequelae of COVID-19 (PASC), are increasingly being recognized as a significant public health challenge [1,2]. Patients with Long COVID experience persistent symptoms, such as fatigue, dyspnea, cognitive dysfunction, and sensory disturbances, often leading to decreased quality of life and productivity [3–5].

Despite the growing awareness regarding Long COVID, its pathophysiology remains debated, and effective preventive and therapeutic strategies remain lacking [6]. Preliminary evidence suggests that early antiviral treatment may reduce the risk of developing Long COVID [7,8]; however, data from randomized controlled trials are limited.

Ensitrelvir fumarate is a novel oral antiviral agent that selectively inhibits the SARS-CoV-2 3C-like protease, a key enzyme in viral replication. In vitro studies have shown that ensitrelvir binds non-covalently to the active site of 3CL protease and blocks viral polyprotein cleavage, thereby suppressing viral replication across various SARS-CoV-2 variants [9]. In a phase 1 trial, ensitrelvir was well tolerated up to 750 mg in healthy adults. The most common adverse event was a transient decrease in HDL cholesterol. The drug showed a long half-life of around 51 hours and no significant effect of food on absorption [10]. It has been granted regulatory approval in Japan for COVID-19 treatment (125 mg tablets; dosage and administration: 375 mg on day 1 and 125 mg on days 2–5) (Shionogi & Co. Ltd, 2024). Ensitrelvir treatment demonstrated decreased viral load compared to placebo in phases 2a and 2b and showed improvements in four respiratory symptoms (stuffy or runny nose, sore throat, shortness of breath, and cough) and as well as the composite of these four respiratory symptoms and feverishness in phase 2b. [11,12]. Although exploratory analyses from the phase 3 SCORPIO-SR trial suggested that early treatment with ensitrelvir may reduce the risk of Long COVID, these findings were not confirmed as part of the pre-specified endpoint of the clinical trial.

Therefore, this study aims to evaluate the efficacy and safety of ensitrelvir fumarate in preventing Long COVID symptoms in patients with mild COVID-19. We hypothesize that early administration of ensitrelvir will reduce the incidence of

persistent symptoms and enhance patient-reported outcomes. Our findings could have important implications for patient care and public health strategies in managing the long-term effects of COVID-19.

## Methods

### Study design and setting

This multi-center, randomized, double-blind, placebo-controlled, parallel-group trial is led by The University of Osaka Hospital, Japan, and conducted at two sites. This study uses a decentralized clinical trial (DCT) approach, with participants recruited from partner healthcare institutions.

### Eligibility criteria

**Inclusion criteria.** Patients eligible for the study include those diagnosed with SARS-CoV-2 infection at a partner site through nucleic acid testing, such as reverse-transcriptase PCR, Loop-mediated Isothermal Amplification or antigen testing. In addition, patients must have mild COVID-19 infection as per severity classification defined by the COVID-19 Medical Treatment Guide Version 10.1 [13]. They should commence the study drug within 72 hours of COVID-19 onset and exhibit pyrexia (temperature ≥ 37°C) at diagnosis. Female patients with childbearing potential must agree to use effective contraception from enrollment to 14 days post-last dose of the study drug. Furthermore, patients must be at least 18 years old and able to provide informed consent via an electronic device.

**Exclusion criteria.** Patients at high risk of developing severe COVID-19 are excluded, as are those who have recently administered with specific drugs for COVID-19, including antiviral drugs (e.g., remdesivir, molnupiravir, nirmatrelvir/ritonavir, and ensitrelvir fumaric acid) and anti-SARS-CoV-2 monoclonal antibodies (e.g., casirivimab and imdevimab, sotrovimab, tixagevimab, and cilgavimab) within 15 days prior to providing informed consent. Additionally, patients who have received humanized anti-interleukin-6 receptor (IL-6R) monoclonal antibodies (e.g., tocilizumab), Janus kinase inhibitors (e.g., baricitinib), or corticosteroids (oral, injection) within 2 weeks prior to consent are also excluded. Participation in a clinical trial for COVID-19 treatment drugs after the onset of the current COVID-19 episode, inability to conduct online interviews using a cell phone with the investigator/sub-investigator, or inability to access the electronic patient-reported outcome system via a cell phone are grounds for exclusion. Patients with allergy/sensitivity or hypersensitivity to components of ensitrelvir, those currently taking contraindicated drugs with ensitrelvir fumaric acid or scheduled to take contraindicated drugs during the treatment period or within 2 weeks after the last dose of the study drug, patients with renal or hepatic dysfunction receiving colchicine, pregnant patients, breastfeeding patients, patients with severe hepatic dysfunction, patients with immunocompromised conditions or undergoing hemodialysis, patients with other infectious diseases, patients participating in other interventional studies or who previously participated in this study, patients who have participated in drug clinical trials within the past year, and patients deemed ineligible for the study by the investigator/sub-investigator for any other reason are also excluded.

### Interventions

Participants will be randomly assigned in a 1:1 ratio to receive either ensitrelvir fumarate (375 mg on day one, followed by 125 mg once daily from days 2–5) or matching placebo. Randomization will be stratified according to COVID-19 vaccination status and severity of COVID-19 symptoms at the time of enrollment. An independent statistician will generate the allocation sequence, which will be implemented through an interactive web response system. Discontinuation criteria include serious adverse events, participant request, investigator's decision, pregnancy, need for prohibited medications, and noncompliance. Adherence will be monitored through pill counts, participant diaries, and telemedicine visits. During the trial, participants may receive medications for COVID-19 symptoms and other conditions at the discretion of the investigator. However, other COVID-19 therapies, immunomodulators, and strong CYP3A inhibitors/inducers will be prohibited.

## Outcomes

The primary efficacy endpoint would be the proportion of patients with either "symptoms of fatigue, shortness of breath, difficulty breathing, disturbances in smell or taste at consecutive 1 and 3 months after the start of treatment" or "symptoms of difficulty in concentrating and thinking, difficulty of reasoning and solving problems, or memory loss (short or long term) at 3 months after the start of treatment." based on self-assessment. The endpoint was defined based on the most common symptoms associated with post COVID-19 condition defined by WHO [14] and the results of the SCORPIO-SR study [15]. Briefly, to exclude non-specific transient symptoms unrelated to Long COVID, we defined Long COVID as at least one symptom among fatigue, shortness of breath and smell taste disorder lasting two consecutive timepoints (1 month and 3 months) or at least one neurological symptom among symptoms of difficulty in concentrating and thinking, difficulty of reasoning and solving problems, or memory loss (short or long term) on 3 months. The symptoms will be defined according to the Long COVID criteria, as outlined in the PCC (Post-COVID Conditions) questionnaire, which standardizes symptom reporting for patients experiencing prolonged effects following COVID-19 [5].

## Key secondary endpoints

Key secondary endpoints of the study encompass several assessments. These include determining the proportion of patients exhibiting symptoms, such as fatigue, shortness of breath, difficulty breathing, smell disturbance, or taste disturbance at consecutive intervals of 1, 3, and 6 months post-treatment initiation, or symptoms of difficulty in concentrating and thinking, arduousness in reasoning and problem-solving, or memory loss (short or long-term) at the 6-month mark. In addition to these, for many of the symptom-based endpoints, a separate analysis will be conducted based on symptoms that participants themselves perceive as being "related" or "possibly related" to their COVID-19 infection, providing a patient-centered perspective on the intervention's efficacy. Symptoms marked as "Yes" or "Not sure" to the question "Symptom: COVID-19 Related?" in the PCC questionnaire are regarded as COVID-19–related or possibly related. In addition, the study strives to determine the proportion of patients experiencing symptoms of fatigue, shortness of breath, difficulty breathing, smell disturbance, or taste disturbance at consecutive 1 and 3-month intervals after treatment initiation. Furthermore, it seeks to ascertain the proportion of patients displaying symptoms of difficulty in concentrating and thinking, difficulty of reasoning and solving problems, or memory loss (short or long term) at the 3-month mark, as well as those with associated neurological symptoms such as insomnia. Finally, it assesses the proportion of patients failing to return to their pre-COVID-19 health status and experiencing any of the specified 14 COVID-19 symptoms at 3 months post-treatment, along with those experiencing any of the listed symptoms at the same interval. Safety will be assessed through monitoring of adverse events and vital signs, physical examinations. Adverse events will be coded using the MedDRA and summarized by treatment group, severity, and relationship with the study treatment.

## Participant timeline

Timeline of the participants, including the schedules of enrollment, interventions, assessments, and visits, is presented in Fig 1.

## Sample size

The sample size was initially calculated based on a test for the difference between two proportions. The assumptions for the calculation were based on data from prior clinical trials of ensitrelvir. Specifically, the event rate for the primary endpoint in the placebo group was estimated to be 14.6%, based on an exploratory analysis of the Japanese population in the SCORPIO-SR trial [10]. We hypothesized that ensitrelvir would reduce the relative risk by 30% (a relative risk of 0.70). With a two-sided alpha level of 5% and 80% power, the required sample size was calculated to be 1,784 participants for the final analysis. Assuming a dropout rate of approximately 10%, a total of 2,000 participants (1,000 per group) will be enrolled.

| | Pre-monitoring period | Medical treatment period | | Follow-up period | | | | | | |
| --- | --- | --- | --- | --- | --- | --- | --- | --- | --- | --- |
| | Day of enrollment | Start date of study medication | 1 week later*2 | After 1 month | After 2 months | After 3 months | After 4 months | After 5 months | After 6 months | Time of discontinuance |
| **Visit** | 1 | - | 2 | 3 | 4 | 5 | 6 | 7 | 8 | |
| **Day** | | 0 | 7 | 28 | 56 | 84 | 112 | 140 | 168 | |
| **Tolerance level** | −*1 | — | −2 to +5 | ±7 | ±7 | ±7 | ±7 | ±7 | ±7 | −*1 |
| **Obtaining Consent** | X | | | | | | | | | |
| **Enrollment and allocation** | X | | | | | | | | | |
| **Background of study participants** | X | | | | | | | | | |
| **Height and weight** | X | | | | | | | | | |
| **Comorbidities** | X | | | | | | | | | |
| **Pre-existing diseases** | X | | | | | | | | | |
| **Study medication initiation** | | X | | | | | | | | |
| **Study medication record** | | ← → | | | | | | | | |
| **Online interview** | X | | X | | | | | | | |
| **Confirmation of concomitant medications** | X | | X | | | | | | | |
| **Pregnancy confirmation** | X | | | | | | | | | |
| **Post-illness symptoms** | | | | X | X | X | X | X | X | (X) |
| **Quality of life** | | X | | X | X | X | X | X | X | (X) |
| **Labor productivity** | | X | | X | X | X | X | X | X | (X) |
| **Medical visits and drug prescriptions** | | | | | X | X | X | X | X | (X) |
| **Adverse events*3** | | | | | | | | | | |
| **Online interview** | X | | X | | | | | | | |
| **Confirmation of concomitant medications** | X | | X | | | | | | | |
| **Pregnancy confirmation** | X | | | | | | | | | |
| **Post-illness symptoms** | | | | X | X | X | X | X | X | (X) |
| **Quality of life** | | X | | X | X | X | X | X | X | (X) |
| **Labor productivity** | | X | | X | X | X | X | X | X | (X) |
| **Medical visits and drug prescriptions** | | | | | X | X | X | X | X | (X) |
| **Adverse events*3** | | ← → | | | | | | | | |

**Fig 1. SPIRIT figure for the schedule of enrolment as recommended by 2013 SPIRIT statement.** *1: Measures should be taken to ensure that the study medication can be started within 72 h of symptom onset. *2: Whenever possible, an online interview will be conducted if the study medication is discontinued, or the study is terminated. *3: Collection of adverse events will begin on day 0 and continue until two weeks after the last dose of study medication.

## Recruitment

Participants will be recruited from partner healthcare institutions collaborating with The University of Osaka Hospital. Physicians at these institutions will identify potential participants and provide them with study information. Interested individuals will be referred to the study team at The University of Osaka Hospital for screening and enrollment. The study was initiated on 16/02/2024 with the recruitment period expected to last 22 months. Data collection, including the 6-month

follow-up period for the last enrolled participants, is projected to be completed by 30/06/2026. The study Completion will be anticipated by December 2026.

## Allocation

An independent statistician will generate the allocation sequence using permuted blocks method. Randomization will be stratified by two factors: history of COVID-19 vaccination (yes/no) and the baseline severity score of 14 COVID-19 symptoms (less than 9 vs. 9 or more). The allocation sequence will be implemented through an interactive web response system and will remain concealed from participants, investigators, and all study staff until the final database lock.

## Blinding

The study participants, investigators, care providers, outcome assessors, and data analysts will be blinded to treatment allocation. The study drugs (ensitrelvir and placebo) will be indistinguishable in appearance, taste, and smell. In case of an emergency, unblinding will be permitted if knowledge of the treatment allocation is essential for the participant's care. The principal investigator will make the decision to unblind, and the event will be documented.

## Data collection methods

This study employs a combination of remote data collection methods to assess and collect outcomes, baseline characteristics, and other trial data. Several processes will be implemented to ensure data quality. Telemedicine visits will be conducted using the "MiROHA" system, which allows secure, high-quality video consultations. Investigators and sub investigators will receive training on the proper use of the system and standardized assessment of participants. Patient-reported outcomes will be collected using the validated "Study Concierge" smartphone application, which features built-in data validation checks and prompts to minimize missing or inconsistent data. Participants will receive clear instructions and reminders to ensure timely completion of assessments. For patient randomization, the study will utilize the Captool system. The data collected through MiROHA and Captool will serve as source documents and will be extracted to create electronic case report forms (eCRFs), which will be designed with input from clinical experts and pretested to ensure clarity, completeness, and ease of use. The study will utilize instruments with established reliability and validity, such as the EuroQol 5-dimension 5-level scale (EQ-5D-5L) and Work Productivity and Activity Impairment Questionnaire, in their Japanese versions to ensure cultural appropriateness. Data entered into the "MiROHA" system and "Study Concierge" application will be automatically transferred to a secure, centralized database, and data quality will be monitored through regular data reviews, data queries, and source data verification.

The study team will employ several strategies to promote participant retention and complete the follow-up. During the process of obtaining informed consent, the participants will be provided clear information regarding the study timeline, procedures, and expectations, with an emphasis on the importance of completing all study assessments. The use of telemedicine visits and smartphone-based data collection is expected to minimize participant burden and facilitate convenient participation from home. The "Study Concierge" application will send automated reminders to the participants for completion of assessments at each time point, and the study team will monitor compliance and follow up with participants as needed. To maintain engagement and reinforce the importance of continued participation, the participants will receive regular study updates and newsletters. For participants who discontinue the intervention or deviate from the protocol, efforts will be made to collect as much outcome data as possible, including the reason for discontinuation or deviation, adverse events, Long COVID symptoms (if available), and EQ-5D-5L and Work Productivity and Activity Impairment Questionnaire responses (if available).

## Data management

Data will be managed according to the Data Management Plan, for data entry, coding, security, and storage. An electronic data capture system will be used with access restricted to authorized personnel. Data validation checks and audit trails will also be performed.

## Statistical methods

The primary endpoint will be analyzed using a modified Poisson regression model, adjusting for stratification factors (COVID-19 vaccination history and baseline symptom score category) and baseline covariates (age, sex, and Body Mass Index [BMI]). Secondary and exploratory endpoints will be analyzed using a similar framework, for which no adjustments for multiplicity. Specifically, for endpoints assessing participant-perceived causality, the presence of symptoms deemed "related" or "possibly related" to COVID-19 will be the response variable in the respective models. Subgroup analyses will be performed based on the stratification factors, as well as age, sex, BMI, history of prior COVID-19 infection, and time from symptom onset to treatment. An interim analysis for overwhelming efficacy, with multiplicity adjustment using an O'Brien-Fleming alpha-spending function, will be conducted upon reaching approximately 50% enrollment. The Full Analysis Set (FAS), defined as all randomized participants who receive at least one dose of the study drug and have at least one post-baseline efficacy assessment, and the Safety Analysis Set (SAS), including all participants who receive at least one dose, will be utilized for efficacy and safety evaluations, respectively. Analyses on the FAS will be based on the intention-to-treat principle, while analyses on the SAS will be based on the treatment received. For the primary analysis, missing data will not be imputed; the analysis will be based on available cases.

## Data monitoring

An independent data-monitoring committee will oversee be established to conduct a single interim analysis for the primary efficacy endpoint. The purpose of this interim analysis is to stop the trial early for overwhelming efficacy. This analysis will be conducted when 909 participants have been enrolled in the Full Analysis Set (FAS), which corresponds to approximately 50% of the total required information for the final analysis. To control the overall type I error rate at a two-sided 0.05 level, the O'Brien-Fleming alpha-spending function will be used to determine the statistical boundary for stopping. The committee will review unblinded interim analysis results and recommend to the trial steering committee whether to continue or terminate the trial.

## Harms

Adverse events (AEs) data will be collected from the initiation of the study drug administration and will continue until 14 days after the final dose of the study drug. The AEs will be assessed for severity (mild, moderate, or severe) and causality (related or unrelated). Serious AEs (SAEs) are defined as any untoward medical occurrence that results in death, is life-threatening, requires inpatient hospitalization or prolongation of existing hospitalization, leads to persistent or significant disability/incapacity, or is a congenital anomaly/birth defect. SAEs will be reported to the sponsor and ethics committee within 7 days of awareness for fatal or life-threatening events and within 15 days for other SAEs. All AEs and SAEs will be followed-up until resolution or stabilization.

## Auditing

The trial will be audited periodically by an independent quality assurance team for evaluation of protocol compliance.

## Ethics and dissemination

The study protocol has been approved by the The University of Osaka Clinical Research Review Committee (CRB5180007, February 2024) and will be conducted in accordance with the Declaration of Helsinki, the Clinical Trials Act, and other relevant regulations. Any significant protocol changes will be submitted as amendments for approval and communicated to relevant parties. Written informed consent will be obtained from all the participants, and their personal information will be kept confidential and made accessible only to the authorized personnel. The principal investigator and sub investigators will declare any potential conflicts of interest, and the sponsor, Shionogi & Co., Ltd.,

will not be involved in data collection, analysis, or interpretation. The principal investigator will have full access to the final dataset, and post-trial data sharing will be considered on a case-by-case basis. Participants who suffer harm from the study will receive appropriate medical treatment and compensation, and after the trial, they will continue to receive standard care. Study results will be disseminated through peer-reviewed publications, conferences, and registries.

## Discussion

This randomized, double-blind, placebo-controlled trial will provide crucial evidence regarding the efficacy and safety of ensitrelvir fumarate in preventing Long COVID symptoms in patients with mild COVID-19. By leveraging the antiviral activity of ensitrelvir and its favorable safety profile, we hypothesize that early treatment initiation may reduce the incidence of persistent symptoms and improve patient-reported outcomes. This study employs a DCT approach, wherein participants will be recruited from partner healthcare institutions and followed up via remote data collection tools. This design offers several advantages, including increased patient diversity and representativeness, reduced participant burden, potentially faster recruitment, and improved retention rates. The DCT model is particularly well-suited for studying Long COVID, as it allows long-term follow-up of participants without requiring frequent in-person visits, which is important in the context of the ongoing pandemic. Our approach aligns with the growing trend of using DCTs to evaluate repurposed medicines for COVID-19, as exemplified by the ACTIV-6 study, which successfully operationalized a decentralized, outpatient randomized platform trial [16].

However, this study has limitations. First, it relies heavily on patient-reported outcomes, which may be subject to recall error. Although patient-reported outcomes are valuable in capturing the subjective experience of Long COVID, the accuracy of these reports may be influenced by various factors, such as time elapsed since symptom onset, severity of symptoms, and overall health status of the participants. The use of objective measures, such as cognitive function tests or biomarkers, could complement patient-reported outcomes and provide a more comprehensive assessment of Long COVID. Second, the study population is limited to patients with mild COVID-19, and the findings may not be generalizable to those with more severe disease. The pathophysiology and risk factors for Long COVID may differ between mild and severe cases, and the efficacy of ensitrelvir in preventing persistent symptoms of severe COVID-19 remains to be determined. Future studies should investigate the potential of antiviral therapy in preventing Long COVID across the spectrum of COVID-19 severity. Third, the DCT approach, which offers several advantages, also presents certain challenges. Ensuring data quality and integrity, maintaining participant engagement, and coordinating with multiple partner institutions may be more difficult in decentralized settings than in traditional site-based trials. The study team has implemented various measures to mitigate these challenges; however, the feasibility and effectiveness of these strategies must be evaluated. Fourth, the study has a relatively short follow-up period of 6 months, which may not capture the full spectrum and duration of Long COVID symptoms. On June 11, 2024, the U.S. National Academies of Sciences, Engineering, and Medicine (NASEM) defined Long COVID as a chronic infection-related condition characterized by persistent, recurrent, remitting, or progressive symptoms affecting one or more organ systems for at least three months. Given this definition, longer follow-up periods, such as 12 weeks or more, are essential to fully understand the long-term impact of Long COVID. Studies have suggested that Long COVID symptoms can persist for ≥ 12 weeks in some patients [17]. Thus, long-term follow-up studies are needed to assess the long-term effects of ensitrelvir on patient outcomes and the natural history of Long COVID.

Nevertheless, this study represents an important step in evaluating the potential of antiviral therapy in preventing Long COVID. If ensitrelvir is found to be effective, it could have considerable implications for patient care and public health. Early treatment with this oral antiviral agent could reduce the burden of persistent symptoms, improve quality of life, and facilitate the return to normal activities. Moreover, preventing Long COVID could lead to substantial healthcare cost savings and productivity gains.

In conclusion, the RESILIENCE trial combines a decentralized design with rigorous methods to evaluate the efficacy and safety of ensitrelvir fumarate in preventing Long COVID in patients with mild COVID-19. The findings of this study may inform clinical practice guidelines and public health strategies for managing the long-term impact of the COVID-19 pandemic, while also providing valuable insights into the design and conduct of future decentralized trials in various therapeutic areas.

## Supporting information

**S1 File. Study protocol eng.** The full study protocol in English.
(DOCX)

**S2 File. Study protocol.** The original full study protocol in Japanese.
(DOCX)

**S3 File. SPIRIT checklist.**
(DOCX)

## Acknowledgments

We would like to thank the participants, investigators, and study staff who will make this trial possible. We also acknowledge the contributions of the Independent Data Monitoring Committee and the clinical trial office at The University of Osaka Hospital.

## Author contributions

**Conceptualization:** Shungo Yamamoto, Ryuichi Minoda Sada, Kento Asano, Daisuke Onozuka, Shintaro Tanaka, Shogo Miyazawa, Masahiro Kinoshita, Satoshi Kutsuna.

**Data curation:** Daisuke Onozuka.

**Formal analysis:** Daisuke Onozuka.

**Investigation:** Keiji Konishi.

**Methodology:** Shungo Yamamoto, Ryuichi Minoda Sada, Kento Asano, Daisuke Onozuka, Satoshi Kutsuna.

**Project administration:** Keiji Konishi.

**Writing – original draft:** Keiji Konishi.

**Writing – review & editing:** Keiji Konishi, Shungo Yamamoto, Ryuichi Minoda Sada, Kento Asano, Daisuke Onozuka, Shintaro Tanaka, Shogo Miyazawa, Masahiro Kinoshita, Satoshi Kutsuna.

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
