## [Decision Letter · Decision Letter 0]

12 Jun 2025

Dear Dr. Konishi,

Thank you for submitting your manuscript to PLOS ONE. After careful consideration, we feel that it has merit but does not fully meet PLOS ONE’s publication criteria as it currently stands. Therefore, we invite you to submit a revised version of the manuscript that addresses the points raised during the review process.

We look forward to receiving your revised manuscript.

Kind regards,

Kamal Sharma

Academic Editor

PLOS ONE

 [This study is funded by Shionogi & Co., Ltd. The funder was involved in the study design, data analysis, and manuscript preparation, but had no role in data collection or the decision to submit the manuscript for publication.]. 

[Keiji Konishi, Daisuke Onozuka, and Satoshi Kutsuna have received research support from Shionogi & Co., Ltd, Tokyo, Japan. Shungo Yamamoto, Ryuichi Minoda Sada, Kento Asano declare no competing interests. The funders have no role in the design of the study; in the collection, analyses, or interpretation of data; in the writing of the manuscript; or in the decision to publish the results. Shintaro Tanaka, Shogo Miyazawa, and Masahiro Kinoshita are full-time employees of Shionogi & Co., Ltd. and may hold stocks in the company.].

Reviewers' comments:

Reviewer's Responses to Questions

**Comments to the Author**

1. Does the manuscript provide a valid rationale for the proposed study, with clearly identified and justified research questions?

Reviewer #1: Yes

Reviewer #2: Yes

2. Is the protocol technically sound and planned in a manner that will lead to a meaningful outcome and allow testing the stated hypotheses?

Reviewer #1: Yes

Reviewer #2: Yes

3. Is the methodology feasible and described in sufficient detail to allow the work to be replicable?

Reviewer #1: Yes

Reviewer #2: Yes

4. Have the authors described where all data underlying the findings will be made available when the study is complete?

Reviewer #1: Yes

Reviewer #2: No

5. Is the manuscript presented in an intelligible fashion and written in standard English?

Reviewer #1: Yes

Reviewer #2: Yes

You may also provide optional suggestions and comments to authors that they might find helpful in planning their study.

Reviewer #1: The effect size used for the sample size calculation needs to be justified.

For a large sample size, a 10% dropout rate may be too optimistic.

The sample size was calculated using a chi-square test. Why is the primary analysis planned with a Poisson model instead?

Please specify the stratification factors and covariates. Will stratification be used in randomization?

Additional details are needed regarding the subgroup and sensitivity analyses.

What is the objective of the interim analysis? Is any adjustment for multiplicity?

Define the Full Analysis Set (FAS) and Safety Set clearly.

What methods will be used for imputing missing data?

What decision will DMC make for efficacy?

Line 275: Please clarify why the sample size is n = 909 for the interim analysis.

Reviewer #2: Following comments needs to be addressed:

Comment 1: The protocol has been submitted now for publication, but study status shows enrolment begun on Feb. 20, 2020 (https://jrct.mhlw.go.jp/en-latest-detail/jRCTs051230184). Authors are encouraged to provide actual status and ethical approval etc.

Comment 2: Since study is on-going, was there any interim data published.

Comment 3: Proposed Mechanism of action of Ensitrelvir (figure is preferred), a brief write-up of prior phase I / II studies of Ensitrelvir with outcome should be helpful for readers.

Comment 4: Reference 4 and 14 are same.

**Do you want your identity to be public for this peer review?** For information about this choice, including consent withdrawal, please see our Privacy Policy

Reviewer #1: No

Reviewer #2: No

---

## [Author Response · Author response to Decision Letter 1]

16 Sep 2025

Journal Requirements

1. Comment: Please ensure that your manuscript meets PLOS ONE's style requirements, including those for file naming.

Response: Thank you for the guidance. We have carefully reviewed the PLOS ONE style templates and revised our manuscript, including file names, to ensure full compliance with the journal's requirements.

2. Comment: Please state what role the funders took in the study.

Response: Thank you for the comment. To clarify the funder's role and resolve a previous inconsistency, we have revised both the Funding and the Competing interests sections of the manuscript.

3. Comment: Please confirm that this does not alter your adherence to all PLOS ONE policies on sharing data and materials, by including the following statement: "This does not alter our adherence to PLOS ONE policies on sharing data and materials.”

Response: As requested, we have added the statement "This does not alter our adherence to PLOS ONE policies on sharing data and materials" to the Competing interests section.

4. Comment: Your ethics statement should only appear in the Methods section of your manuscript. If your ethics statement is written in any section besides the Methods, please move it to the Methods section and delete it from any other section.

Response: Thank you for this instruction. As requested, we have moved the ethics statement regarding the institutional review board approval to the main Methods section, under the "Study design and setting" subsection. The original section has been renamed to "Dissemination" to accurately reflect its remaining content.

5. Comment: Please include captions for your Supporting Information files at the end of your manuscript...

Response: Our manuscript does not include any Supporting Information files at this time.

Reviewer #1

1. Comment: The effect size used for the sample size calculation needs to be justified.

Response: Thank you for this important point. We have clarified the basis for our sample size calculation in the manuscript. The assumed 14.6% event rate in the placebo group was based on an exploratory analysis of data from the preceding SCORPIO-SR trial. The 30% relative risk reduction (i.e., a relative risk of 0.70) was subsequently established as a clinically meaningful and plausible target for the intervention effect, informed by the potential suggested in the prior trial. This has been clarified in the Sample size section.

2. Comment: For a large sample size, a 10% dropout rate may be too optimistic.

Response: We appreciate the reviewer's valid concern. We believe the 10% rate is a reasonable initial estimate due to the participant-friendly decentralized clinical trial (DCT) design, which is expected to enhance retention by minimizing participant burden. However, we acknowledge this may be optimistic and, as the reviewer implies, a higher-than-expected dropout rate is a potential limitation. We have ensured this point is clearly stated in the Discussion section of the manuscript.

3. Comment: The sample size was calculated using a chi-square test. Why is the primary analysis planned with a Poisson model instead?

Response: This is an excellent question. While a straightforward chi-square test was used for the sample size calculation, we chose a modified Poisson regression model for the primary analysis. This approach allows for the estimation of relative risks while robustly adjusting for stratification factors and key baseline covariates, thereby providing a more precise and powerful estimate of the treatment effect. We have added a sentence to the Statistical methods section to clarify this rationale.

4. Comment: Please specify the stratification factors and covariates. Will stratification be used in randomization?

Response: Thank you for the suggestion. We have revised the manuscript to explicitly state that randomization will be stratified by COVID-19 vaccination history (yes/no) and baseline severity score of 14 COVID-19 symptoms (less than 9 vs. 9 or more). We have also specified in the Statistical methods section that the primary analysis model will adjust for these stratification factors and the baseline covariates of age, sex, and Body Mass Index (BMI).

5. Comment: Additional details are needed regarding the subgroup and sensitivity analyses.

Response: We have expanded the Statistical methods section to provide these details. Subgroup analyses for the primary endpoint will be performed based on the stratification factors, as well as age, sex, BMI, history of prior COVID-19 infection, and time from symptom onset to treatment.

6. Comment: What is the objective of the interim analysis? Is any adjustment for multiplicity?

Response: We have clarified in the Data monitoring section that the objective of the single interim analysis is to stop the trial early for overwhelming efficacy. To control the overall type I error rate, we will use the O'Brien-Fleming alpha-spending function to adjust for multiplicity.

7. Comment: Define the Full Analysis Set (FAS) and Safety Set clearly.

Response: Thank you. We have added clear definitions for the Full Analysis Set (FAS) and the Safety Analysis Set (SAS) to the Statistical methods section.

8. Comment: What methods will be used for imputing missing data?

Response: We have clarified in the Statistical methods section that the primary analysis will be based on available cases without imputation.

9. Comment: What decision will DMC make for efficacy?

Response: We have clarified in the Data monitoring section that the independent Data Monitoring Committee (DMC) will review the unblinded interim analysis results and recommend to the trial steering committee whether to continue or terminate the trial based on the pre-specified statistical stopping boundary for overwhelming efficacy.

10. Comment: Line 275: Please clarify why the sample size is n = 909 for the interim analysis.

Response: We have clarified this point in the Data monitoring section. The interim analysis will be conducted when 909 participants have been enrolled in the Full Analysis Set (FAS). This number corresponds to approximately 50% of the total required statistical information for the final analysis, not 50% of the participants.

Reviewer #2

1. Comment: The protocol has been submitted now for publication, but study status shows enrolment begun on Feb. 20, 2020 (https://jrct.mhlw.go.jp/en-latest-detail/jRCTs051230184). Authors are encouraged to provide actual status and ethical approval etc.

Response: Thank you for your review and for spotting the typo in your comment. The trial began in February 2024, not 2020, which is consistent with the clinical trial registry (jRCT) and the manuscript. Participant recruitment is currently proceeding as planned.

2. Comment: Since study is on-going, was there any interim data published.

Response: As the study is ongoing and remains blinded to investigators and participants, no interim data have been published.

3. Comment: Proposed Mechanism of action of Ensitrelvir (figure is preferred), a brief write-up of prior phase I / II studies of Ensitrelvir with outcome should be helpful for readers.

Response: Thank you for this valuable suggestion. We have significantly expanded the Introduction section to include details on ensitrelvir's mechanism of action (as a 3C-like protease inhibitor) and a summary of key findings from prior Phase I and II/III trials regarding its safety, tolerability, and efficacy on viral load and acute symptoms.

4. Comment: Reference 4 and 14 are same.

Response: We sincerely thank you for identifying this error. We have removed the duplicate reference, renumbered the entire reference list accordingly, and double-checked all citations in the text.

---

## [Decision Letter · Decision Letter 1]

15 Oct 2025

Research to Evaluate Safety and Impact of Long COVID Intervention with Ensitrelvir for National Cohort (RESILIENCE Study): A protocol for a randomized, double-blind, placebo-controlled trial

PONE-D-25-15689R1

Dear Dr. Konishi,

We’re pleased to inform you that your manuscript has been judged scientifically suitable for publication and will be formally accepted for publication once it meets all outstanding technical requirements.

Kind regards,

Kamal Sharma

Academic Editor

PLOS ONE

Additional Editor Comments (optional):

Reviewers' comments:

Reviewer's Responses to Questions

**Comments to the Author**

1. Does the manuscript provide a valid rationale for the proposed study, with clearly identified and justified research questions?

Reviewer #1: Yes

2. Is the protocol technically sound and planned in a manner that will lead to a meaningful outcome and allow testing the stated hypotheses?

Reviewer #1: Yes

3. Is the methodology feasible and described in sufficient detail to allow the work to be replicable?

Reviewer #1: Yes

4. Have the authors described where all data underlying the findings will be made available when the study is complete?

Reviewer #1: Yes

5. Is the manuscript presented in an intelligible fashion and written in standard English?

Reviewer #1: Yes

You may also provide optional suggestions and comments to authors that they might find helpful in planning their study.

Reviewer #1: All my concerns are addressed.

**Do you want your identity to be public for this peer review?** For information about this choice, including consent withdrawal, please see our Privacy Policy

Reviewer #1: No

---

## [Editor Report · Acceptance letter]

PONE-D-25-15689R1

PLOS ONE

Dear Dr. Konishi,

I'm pleased to inform you that your manuscript has been deemed suitable for publication in PLOS ONE. Congratulations! Your manuscript is now being handed over to our production team.

Kind regards,

on behalf of

Dr. Kamal Sharma

Academic Editor

PLOS ONE